# Performance of Rubber Concrete Containing Polypropylene and Basalt Fibers under Coupled Sulfate Attack and Freeze–Thaw Conditions: An Experimental Evaluation

**DOI:** 10.3390/polym15092066

**Published:** 2023-04-26

**Authors:** Tao Ran, Jianyong Pang, Jincheng Yu

**Affiliations:** 1School of Civil Engineering and Architecture, Anhui University of Science and Technology, Huainan 232001, China; 2China Construction Fourth Engineering Bureau Sixth Co., Ltd., Hefei 230011, China

**Keywords:** concrete, rubber, microstructure, polymer fiber

## Abstract

Rubber concrete (RC) is a new type of concrete that is currently receiving a lot of attention, solving serious pollution problems by grinding waste tires into granules and adding them to concrete. However, rubber concrete has deficiencies in mechanics and durability, and has been reinforced by adding fibers in many studies. In this study, the mechanical and durability properties of rubber concrete with added polypropylene and basalt fibers (PBRC) were investigated in a series of experiments including apparent morphology, mass, static compressive and tensile tests, ultrasonic non-destructive testing, and scanning electron microscope (SEM) tests under coupled environments of sulfate attack and freeze–thaw. The results showed that the mass loss rate of RC and PBRC gradually increased with the number of freeze–thaw cycles, with more pits and cement paste peeling from the specimen surface. Moreover, the compressive and splitting tensile strengths of RC and PBRC groups exhibited distinct trends, with the former group showing a lower residual strength relative to the latter. The residual compressive strength of the RC group was only 69.4% after 160 freeze–thaw cycles in 5% MgSO_4_ solution. However, it is worth noting that the addition of too many fibers also had a negative effect on the strength of the rubber concrete. Additionally, the scanning electron microscopy (SEM) results indicated that the fibers restricted the formation of microcracks in the microstructure and curtailed the brittleness of the concrete. This study can provide a valuable reference for the application of environmentally friendly material fibers in recycled aggregate concrete.

## 1. Introduction

Concrete is one of the most widely used building materials in the world and consumes a huge amount of resources every year [1]. At the same time, with the development of the transportation industry and the automobile industry, a large number of rubber tires are discarded and landfilled [2]. It is estimated that nearly 100 billion tires reach the end of their useful life each year, and more than 50% of them are discarded without any treatment [3]. As rubber takes a long time to degrade in nature, the accumulation of these discarded tires is a serious hazard to human health and biodiversity [4]. One of the research hotspots in recent years is the use of waste rubber tires to produce concrete. A feasible solution to utilize waste tire rubber is to incorporate them into concrete to replace natural aggregates such as sand. Grinding tire rubber into a small particle form (crumb rubber) and applying it to cement-based materials such as concrete is a resource-saving and environment-friendly method.

For the construction industry, using rubber to improve the inherent properties of concrete is an effective method. The results show that rubber concrete is superior to ordinary concrete in impact resistance [5,6], chloride ion erosion resistance [7], toughness [8], and durability [9,10]. However, according to previous reports, rubber concrete has a major weakness in terms of strength reduction. It is generally believed that the introduction of rubber will reduce the mechanical properties of concrete [11]. At the same time, the compressive strength and elastic modulus of rubber concrete decrease significantly with increasing rubber content [12,13,14]. Li’s [15] experiments proved that the decrease in the strength of rubber concrete is due to the hydrophobicity and poor adhesion of the rubber itself. According to Park’s [16] report, in order not to significantly reduce the compressive strength of concrete, the rubber content should not exceed 17–20% of the total aggregate.

Therefore, mitigating the reduction in the mechanical properties of concrete can maximize the benefits of adding rubber to concrete. The use of fibers helps to improve the properties of concrete, which can compensate for the reduction in mechanical properties due to the addition of high percentages of rubber [17,18,19]. Wang [20] investigated the effects of PVA fiber and MgO on the abrasion and cracking resistance of hydraulic concrete in hydropower projects. Adding 4–8% MgO reduces strength and abrasion resistance, while 1.2–2.4 kg/m^3^ PVA fiber increases splitting tensile strength and cracking resistance. Recently, researchers have conducted extensive studies on composite fiber and crumb rubber concrete to explore the synergistic effect between them. Alwesabi [21] used a mixture of polypropylene fibers and steel fibers to improve the mechanical properties and sustainability of rubberized concrete. Experimental results show that mixed fibers can significantly improve the fracture characteristics and crack resistance of concrete, and the mixture of 0.1% polypropylene fiber–0.9% steel fiber is the optimal mixing ratio. Alsaif [22] introduced a study on steel fiber-reinforced rubber concrete, which found that adding steel fibers to concrete containing waste tire rubber can significantly alleviate the loss of bending strength (from 50% to 9.6% compared with ordinary concrete). Wang [23] evaluated the synergistic effect of macroscopic polypropylene (PP) fibers combined with rubber concrete, based on the evaluation of mechanical properties, durability, and microstructure. The results showed that macroscopic PP fibers enhanced the durability of rubber concrete, including drying shrinkage and resistance to frost. The combination of rubber concrete and macroscopic PP fibers can enhance residual load capacity, distribute stress, and reduce brittleness.

At the same time, as a building material, concrete durability research cannot be ignored, especially the damage to buildings under freeze–thaw erosion conditions [24,25,26]. Most studies have proved that the salt freezing cycle increases the porosity and average pore size of concrete and reduces the bond strength between aggregates in cement matrix [27,28,29]. In order to improve the durability of concrete under freeze–thaw damage, researchers have undertaken a lot of work. Richardson [30] used polypropylene fiber in concrete, and the fiber content of 0.9 kg/m^3^ significantly improved the frost resistance of concrete without reducing the compressive strength and elastic modulus. Mohajerani [31] believed that when the rubber content in C40 concrete is 5.4%, the anti-freeze and thaw effect is the best. Gesoglu [32] found that the freeze–thaw durability problem of pervious concrete can be solved by using rubber, and demonstrated that small-sized rubber particles exhibit better performance than large-sized rubber particles. Therefore, by mixing rubber particles and fibers, the salt freeze–thaw resistance of concrete can be improved without reducing the mechanical properties of concrete.

This study aims to improve the salt freeze–thaw resistance of rubber concrete by adding polypropylene and basalt mixed fibers. The slow freezing experiment method was used to study the apparent deterioration, mass loss rule, mechanical property change, and loss degree analysis of fiber-reinforced rubber concrete under the sulfate freeze–thaw coupling environment. Finally, the influence of fibers on the properties of concrete at the microscopic level was analyzed by scanning electron microscopy (SEM). This research is helpful to improve the utilization degree of solid waste in concrete, which is of great significance to environmental protection and natural resource conservation.

## 2. Materials and Methods

### 2.1. Raw Materials

Coarse aggregate is continuously graded crushed stone with a particle size of 5–20 mm, and the mud content is less than 0.2%. The fine aggregate is natural river sand with a fineness modulus of 2.52, a bulk density of 1450 kg/m^3^, an apparent density of 2570 kg/m^3^, and a mud content of <1.0%. The particle size curves of coarse and fine aggregates are given in Figure 1. The water reducer used in the test is a polycarboxylate high-performance water reducer, a light yellow, odorless liquid with a density of 1.031 kg/m^3^ and a water-reducing rate of 27.4%. The chemical composition of the cement is given in Table 1. The rubber is 40–60 mesh rubber powder with an apparent density of 1160 kg/m^3^. The basalt fiber (BF) is produced by Shanxi Xuanwu Company, with a length of 18 mm. The polypropylene fiber (PPF) is produced by Xinya Construction Technology Company and is in the shape of a bundled monofilament with a length of 10 mm. The specific image of the fiber is shown in Figure 2, and the main performance parameters of the two fibers are shown in Table 2.

### 2.2. Mix Design

The concrete used in the test adopts the same water–cement ratio. According to the “Common Concrete Mix Proportion Design Regulations”, after many laboratory trials, W/B = 0.383 is finally determined. The concrete foundation mix ratio is given in Table 3. According to the previous literature research, the mixing ratio of BF and PPF is 2:1, so seven different fiber volume dosages are designed, and the content of basalt fiber is from 0% to 0.6%. The fine aggregate (sand) in the concrete is replaced with an equal volume of rubber powder. The replaced volume is called the rubber replacement rate, and the concrete is prepared at a rubber replacement rate of 5%. The specific mixing ratio is shown in Table 4. Each group was further divided into 5 groups (corresponding to 0, 40, 80, 120, and 160 sulfate freeze–thaw cycles). In order to reduce the error, 3 samples were used in each group, a total of 105 samples. The sample size is 100 mm × 100 mm × 100 mm.

### 2.3. Preparation of Specimens

The concrete used in the test was mixed by a horizontal mixer, and the flow chart of the preparation of the concrete specimens is shown in Figure 3. Put the weighed sand, gravel, and rubber powder into the mixing bucket for dry mixing for 2 min, evenly sprinkle the torn fibers in the process, and then stir for 1 min. The interaction force generated during the friction and collision during the dry mixing process breaks up the fibers to ensure that the fibers can be evenly dispersed in the aggregate first. Then, add the gelling material into the mixing bucket, stir for 1 min, and finally pour the pre-mixed water and water reducer into the mixing bucket twice, first pour half of it, start wet mixing for 1 min, and then pour the rest and stir for 3 min. Put the mixed concrete into the mold, vibrate fully, and let it stand for 24 h before demolding. The concrete should be cured in a saturated Ca(OH)_2_ solution at (20 ± 2) °C for 28 days.

### 2.4. Erosion Freeze–Thaw Cycle Program

(1)The concrete specimens cured for 28 days were taken out from the saturated Ca(OH)_2_ solution, rinsed with clean water, grouped, and numbered, and the initial mass and wave velocity of the specimens were measured, both of which should be wiped off with a wrung-out wet towel before starting the subsequent operation, and finally the apparent form of each group of specimens was photographed and recorded.(2)Prepare several water tanks, pour in a certain amount of water, 5% Na_2_SO_4_ solution and 5% MgSO_4_ solution, and install a heating device to ensure that the liquid temperature of the concrete specimens immersed in the water tank during the thawing period is maintained in the range of 18–20 °C, as shown in Figure 4.(3)In the “Slow freezing method” freezing test, using 100 mm × 100 mm × 100 mm cubic specimens, the number of freeze–thaw cycles is set to 40 times, 80 times, 120 times, and 160 times in a turn. Each cycle uses 8 h freezing + 4 h thawing, freezing in the freezing test chamber maintains the temperature in the range of −20~−18 °C, and the freezing and thawing process of temperature conversion control takes 30 min to complete, as shown in Figure 5.(4)After each completion of 20 cycles, the weighing of specimens was carried out. Wave velocity, appearance, and mechanical tests (compressive test and splitting tensile test) were carried out promptly after the completion of the predetermined number of cycles, the damaged specimens were collected after the mechanical tests, and samples were made for SEM tests.

### 2.5. Testing Methods

(1) Compressive test

Concrete cube compressive strength test and splitting tensile strength test, both using the side length of 100 mm cube specimen. The loading apparatus is a WAW-1000 microcomputer-controlled electro-hydraulic servo universal testing machine, using displacement as the speed control index of loading, the loading rate of the compressive strength test is 3 mm/min, and the preload force is set to 500 N; the loading rate of the splitting tensile strength test is 1 mm/min, and the preload force is set to 50 N. The formulae for calculating the compressive strength and splitting tensile strength of concrete are given below.
(1)$$f_{\mathrm{cc}}=\beta \times \frac{F}{A}$$
(2)$$f_{\mathrm{ts}}=\beta \times \frac{2F}{\pi A}=\beta \times 0.637\frac{F}{A}$$

$f_{\mathrm{cc}}$ is the compressive strength of the concrete cube, MPa; *F* is the peak load when the concrete specimen is damaged, N; *A* is the compressive area of the concrete specimen, *A* is the compressive area of the concrete specimen, mm^2^; and β is the size conversion coefficient, taken as 0.95 (1).

$f_{\mathrm{ts}}$—splitting tensile strength, MPa; β—dimensional conversion factor, taken as 0.85; *F*—breaking load, N; *A*—the area of splitting surface, mm^2^ (2).

(2) Non-destructive testing

The damage to the concrete is analyzed by measuring the wave velocity of the specimen before and after the erosion freeze–thaw cycle with a non-metallic ultrasonic testing analyzer. In order to quantify the damage to the concrete specimens after erosion freeze–thaw cycles, the relative wave velocity $V_{r}$ and the damage degree $D_{ni}^{s}$ are defined, and each group is calculated as follows
(3)$$V_{r}=\frac{V_{{n\_}_{i}}^{s}}{V_{0}}$$
(4)$$D_{{n\_}_{i}}^{s}=1-V_{r}$$

*n* is the number of freeze–thaw cycles, $V_{0\_i}^{s}$ is the initial wave velocity of the specimen before the test, km/s; $V_{{n\_}_{i}}^{s}$ is the wave velocity of the specimen after n freeze–thaw cycles km/s; $V_{r}$ is the relative wave velocity before and after the test; $D_{{n\_}_{i}}^{s}$ is the damage degree of the specimen after *n* freeze–thaw cycles.

(3) Mass loss

The mass loss rate of concrete under coupled erosion freeze–thaw action is calculated by Equation (5).
(5)$$∆W_{{n\_}_{i}}^{s}=\frac{m_{{0\_}_{i}}^{s}-m_{{n\_}_{i}}^{s}}{m_{{n\_}_{i}}^{s}}\times 100\%$$

*n* is the number of freeze–thaw cycles, $\Delta W_{{n\_}_{i}}^{s}$ is the mass loss rate of the specimen after n freeze–thaw cycles, %; $m_{{0\_}_{i}}^{s}$ is the initial mass of the specimen before the test, g; $m_{{n\_}_{i}}^{s}$ is the mass of the specimen after n freeze–thaw cycles, g.

## 3. Results and Discussion

### 3.1. Damage Analysis of Concrete Apparent Morphology

Due to the dual erosive effect of magnesium and sulfate ions in the MgSO_4_ solution on the concrete, the most severe apparent damage was observed for the same specimens in the 5% MgSO_4_ solution, while the damage in clear water and 5% Na_2_SO_4_ solution was not significant. In order to show more visually the changes in the appearance of the concrete and to record the damage between every 40 freeze–thaw cycles, the changes in the appearance of the concrete specimens in 5% MgSO_4_ solution are given in Figure 6.

It can be observed in the figure that after 40 freeze–thaw cycles, the concrete specimens in 5% MgSO_4_ solution show pits of various sizes on the surface and a small amount of cement paste spalling at local locations, and that the erosion effect has become more pronounced with increasing fiber admixture for the three groups of specimens after P3B3. After 80 freeze–thaw cycles, the cement paste spalling on the concrete surface has increased, while the surface has become rough due to the exposure of sand particles, the number of pits has increased, and the volume of the original pits has increased, and especially on the upper surface of the specimen, more deep pits have been formed, but the coarse aggregate has not been exposed. After 120 freeze–thaw cycles, the surface of the concrete specimen has become rougher, the mortar particles have flaked significantly, even a small amount of coarse aggregate has been exposed and has slightly dropped, and the fibers have been gradually revealed. After 160 freeze–thaw cycles, the surface cement paste of the concrete specimens has produced a large area of spalling, the phenomenon of exposed coarse aggregate is significant, and the situation of falling corners has increased. However, at this time, the P3B3 group’s apparent form is still relatively intact; when the mixed fiber dosing is low, the specimen surface cement mortar layer spalling is more serious, while the high dose specimen surface pits show a serious situation.

Comprehensive analysis of the concrete surface damage and the incorporation of fibers shows that when the total amount of fibers is not large, it can effectively delay the surface cement paste and fine aggregate spalling, so that the apparent form of the test specimens remains more complete, including the P3B3 group. When the total amount of fibers is too large, the surface of the specimen is damaged more seriously and will be damaged prematurely.

### 3.2. Concrete Mass Loss Law

Since the mass loss rate changes among R5, P1B1, and P4B4; P2B2 and P3B3; and P5B5 and P6B6 were all relatively similar, the three groups with the greatest variability were specifically selected as R5, P3B3, and P6B6, respectively, to plot the respective mass loss change curves under the action of freeze–thaw cycles in clear water, 5% Na_2_SO_4_ solution, and 5% MgSO_4_ solution, as shown in Figure 7, Figure 8 and Figure 9.

It can be observed from the figure that the mass loss rate of each group of specimens continues to rise as the number of freeze–thaw cycles increases. The mass loss is mainly due to the specimens being filled with water; when the internal pores are filled with water, and the temperature drops, the water freezes into ice and the volume expands to produce the freezing and swelling force, and with the repeated freeze–thawing, the cement paste on the surface of the specimens together with the sand and gravel aggregates begin to gradually fall off. In all three solutions, the mass loss rate of the P3B3 group was significantly lower than the other groups. After completing 160 freeze–thaw erosions in 5% MgSO_4_, the most severe erosion effect, the mass loss rates of R5, P3B3, and P6B6 in 5% MgSO_4_ were 2.62%, 2.04%, and 2.9%, respectively, and the P3B3 group was reduced by 28.4% and 42.1%, respectively. At the right admixture, the incorporated fibers make the concrete denser and less porous and due to the lack of open pores, the salt solution only penetrates in a shallow range below the concrete surface, and the content decays rapidly when the erosion ions enter the concrete interior. However, the agglomeration of fibers at large dosing levels and the agglomeration of polypropylene fibers float very easily, resulting in more severe damage to P6B6 compared to R5. Additionally, the mass loss rate of P3B3 is higher than in the experiments in clear water and 5% NaSO_4_ due to the double erosion of SO_4_^2−^ and Mg^2+^ in the MgSO_4_ solution.

### 3.3. Compressive Strength

From Figure 10a,b, it can be seen that the compressive strength of concrete specimens under the action of water freeze–thaw cycles decreases continuously with the increase in the number of freeze–thaw cycles, the decline of compressive strength is relatively small at the early stage of freeze–thaw cycles, and the residual rate of compressive strength of each specimen at 40 freeze–thaw cycles is greater than 95%. In particular, the final residual strength of the R5 group was only 78.65%, which was the lowest value among all groups.

Figure 10c,e show the changes of compressive strength of the specimens in each group after freeze–thaw cycles in 5% Na_2_SO_4_ solution and 5% MgSO_4_ solution. Unlike in water, in 5% Na_2_SO_4_ solution and 5% MgSO_4_ solution, the concrete strength increased in the first period of the freeze–thaw cycle, which is due to the reaction between sulfate solution and cement hydration products to generate calcium alumina to fill some of the pores inside the concrete, while the hydration reaction in the concrete still continues, both of which greatly improve the compactness of the concrete structure, while the freeze–thaw damage at this time is relatively small, which is manifested as an increase in compressive strength macroscopically. After 160 freeze–thaw cycles, the compressive strength of the reference group R5 in water, 5% Na_2_SO_4_ solution, and 5% MgSO_4_ solution decreased by 21.35%, 24.76%, and 30.44%, respectively, with the most severe loss of compressive strength in 5% MgSO_4_ solution and the lowest rate of compressive strength loss in water. The P4B4 group had the highest residual rate of compressive strength after freeze–thaw cycling in both 5% Na_2_SO_4_ solution and 5% MgSO_4_ solution, reaching 83.49% and 80.16%, respectively.

Freeze–thaw cycles and sulfate attack will lead to the formation and expansion of cracks, and the most important function of the fibers is to stop the cracks. The disorderly distribution of fibers and concrete matrix are closely bonded to play a suppressive role in the formation and expansion of cracks, while the fiber “bridging” role can also limit the expansion of pores, so incorporation of the fiber into concrete can effectively mitigate damage due to sulfate erosion and freeze–thaw cycles, delaying the damage to the concrete. However, in large amounts, the frost resistance and corrosion resistance of concrete are reduced, due to the fact that the positive effect of the fiber no longer exists, while the negative effects (such as fiber agglomeration) will increase the initial defects of concrete, aggravating the damage caused by sulfate attack and freeze–thaw cycles on the concrete.

### 3.4. Splitting Tensile Strength

As can be seen from Figure 11, the splitting tensile strength of concrete specimens under the action of clear water freeze–thaw cycles decreased continuously with the increase in the number of cycles, and the loss rate of splitting tensile strength did not exceed 5% at 40 freeze–thaw cycles, where the decrease rate of R5 was the largest, which was 3.86%. After 160 freeze–thaw cycles, the splitting tensile strength loss rate of each test group was as follows: R5 (26.38%) > P2B2 (21.26%) > P1B1 (19.3%) > P6B6 (18.92%) > P5B5 (16.93%) > P3B3 (14.57%) > P4B4 (12.91%); the highest residual splitting tensile strength of the specimens was found in P4B4 with 3.45 MPa; R5 was the lowest with only 2.38 MPa.

The splitting tensile strength patterns of concrete specimens in 5% Na_2_SO_4_ solution and 5% MgSO_4_ solution were similar. The tensile strength of concrete with suitable fiber content showed a significant increase, and the strength of P2B2 and P3B3 was higher than the other experimental groups, especially the control group R5, in the early and late stages of freeze–thaw cycles. After 160 freeze–thaw cycles in 5% MgSO_4_ solution, P2B2 and P3B3 improved by 69.7% and 56.7%, respectively, compared with the control group. In addition, under the condition of large fiber content, the splitting strength of the experimental group was still higher than the control group without fiber after 160 freeze–thaw cycles, which proved that the improvement of concrete splitting strength was significant with fiber. The addition of mixed fibers in concrete can play a role in transmitting force and absorbing energy, thus improving the tensile properties of concrete and avoiding stress concentration. Sulfate erosion and freeze–thaw cycles continue to produce many microcracks inside the concrete, and mixed fibers, due to their greater tensile strength and modulus of elasticity, can be bridged at the crack inhibition, crack generation, and expansion to improve the tensile strength and crack resistance of concrete.

### 3.5. Analysis of the Degree of Damage

Figure 12 shows the damage degree of each group of specimens after completing 160 freeze–thaw cycles in water, 5% Na_2_SO_4_ solution, and 5% MgSO_4_ solution. For the same specimens, the highest damage was found in 5% MgSO_4_ solution, followed by 5% Na_2_SO_4_ solution, and the lowest in water; for the same solutions, the specimens with the highest and lowest damage in water were R5 and P3B3, the specimens with the highest and lowest damage in 5% Na_2_SO_4_ solution were P6B6 and P3B3. The specimens with the smallest damage in 5% MgSO_4_ solution were P6B6 and P4B4, which roughly corresponded to the pattern of compressive strength loss and splitting tensile strength loss after 160 cycles. The P3B3 group with the smallest degree of loss had a loss of 0.058 after 160 freeze–thaw cycles in water, which was 62.3% lower than that of the R5 group (0.154). The addition of an appropriate amount of fibers can reduce the damage degree of concrete after experiencing coupled erosion freeze–thaw cycles, but it is worth noting that the excessive use of mixed fibers can also increase the loss degree. Taken together, group P3B3 demonstrates the optimum amount of mixed fibers to be added.

### 3.6. Microscopic Electron Microscope Analysis

The microstructure of the concrete was observed by scanning electron microscopy, and the images obtained are given in Figure 13. Several cracks are created in the concrete matrix, and the fibers span across the cracks before transferring the stresses and transferring the load from one side to the other (Figure 13a,b); the cracks in the concrete matrix encounter the fibers and are prevented from expanding by them, i.e., the “crack arresting” effect. The grip force exerted by the cement paste on the fibers helps the fibers to play the role of “crack arresting”, and due to the anchoring effect of the cement paste on the fibers, the fibers can transfer the load they bear to the adjacent cement matrix, avoiding the occurrence of stress concentration (Figure 13c,d). The inhomogeneous dispersion of fibers inside the concrete is shown in Figure 13e,f. The fibers are distributed in bundles next to each other, and the gaps existing between them cannot be completely filled by the cement matrix due to the close spacing of the fibers, and sufficient contact between the fibers and the cement matrix cannot be obtained. When the concrete is loaded, this part of the cement paste will be rapidly damaged and the fibers will be peeled off and will fail due to the loss of the cement paste package, which will become the weak zone of the concrete matrix and lead to the deterioration of the concrete performance.

## 4. Conclusions

In this study, the effects of the number of freeze–thaw cycles, the type of erosion solution, and the mixed fiber content on the mass loss rate, apparent morphology, relative wave velocity, and mechanical properties of rubber concrete were investigated. In contrast to previous studies, different solution types and fiber types were mentioned. The following conclusions can be drawn.

(1)For the same specimen, under a certain number of freeze–thaw cycles, the apparent deterioration is most serious in 5% MgSO_4_ solution, and the deterioration in clear water and 5% Na_2_SO4 solution is similar. Appropriate dosing of fibers can effectively prolong the occurrence of spalling in cement paste and fine aggregate on the surface. Nevertheless, an excessive total fiber dosage can aggravate surface damage of the specimens, causing premature failure.(2)With the increase in the number of freeze–thaw cycles, the mass loss rate of the specimens in each group showed an overall increasing trend, with a maximum of 2.87%. The mass loss rate of the P3B3 group was significantly lower than that of the other groups, which was only 2.03%. Adding fibers can increase the compactness of concrete, but when a large amount is added, the fibers agglomerate easily, resulting in quality loss.(3)The strength of the concrete decreased with the increase in the number of cycles, but the rubber concrete mixed with 0.2% BF and 0.1% PPF showed better residual compressive strength and splitting tensile strength due to the bridging effect and crack arresting effect of the mixed fibers 41.3 Mpa and 3.26 Mpa, respectively. However, with the continuous increase in fiber content, the mechanical properties showed the same deterioration trend.(4)The damage degree and strength loss rate of each specimen are relatively similar, and the damage degree under high fiber content is not decreasing but increasing compared with the reference group, and the damage degree is smaller with suitable fiber content. The degree of damage to concrete has a greater impact on the mechanical properties after freeze–thaw cycles, and the greater the degree of damage, the higher the strength loss.(5)The microstructure of the specimens after salt freeze–thaw cycles was observed by SEM. Fibers play a bridging role in the microstructure, which can prevent crack expansion and stress concentration. The contact between the fibers and the cement matrix is critical, and voids between the fiber bundles and the cement matrix can lead to fiber failure and affect the concrete performance.

## Figures and Tables

**Figure 1 polymers-15-02066-f001:**
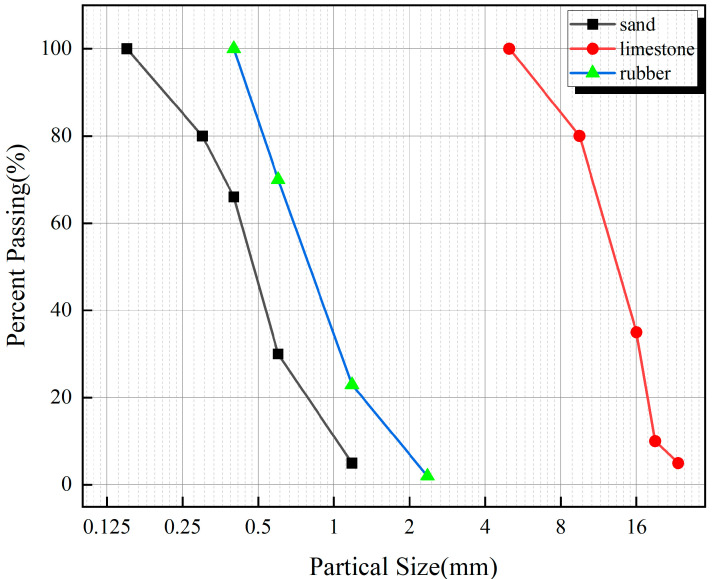
Coarse and fine aggregate particle size curve.

**Figure 2 polymers-15-02066-f002:**
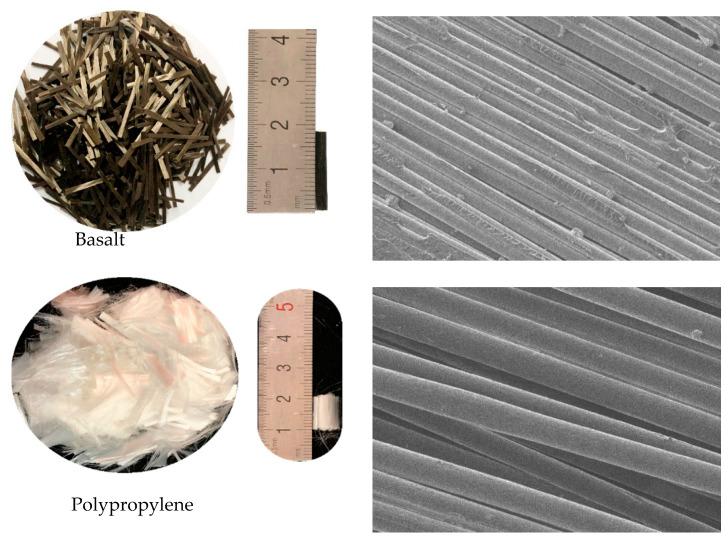
Fiber pictures and microscopic images.

**Figure 3 polymers-15-02066-f003:**
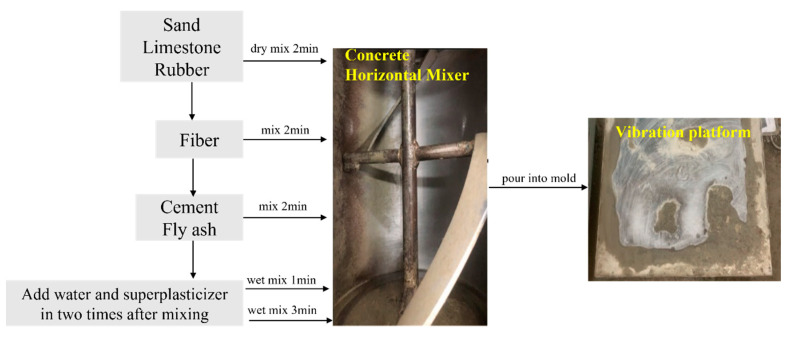
Concrete Preparation Process.

**Figure 4 polymers-15-02066-f004:**
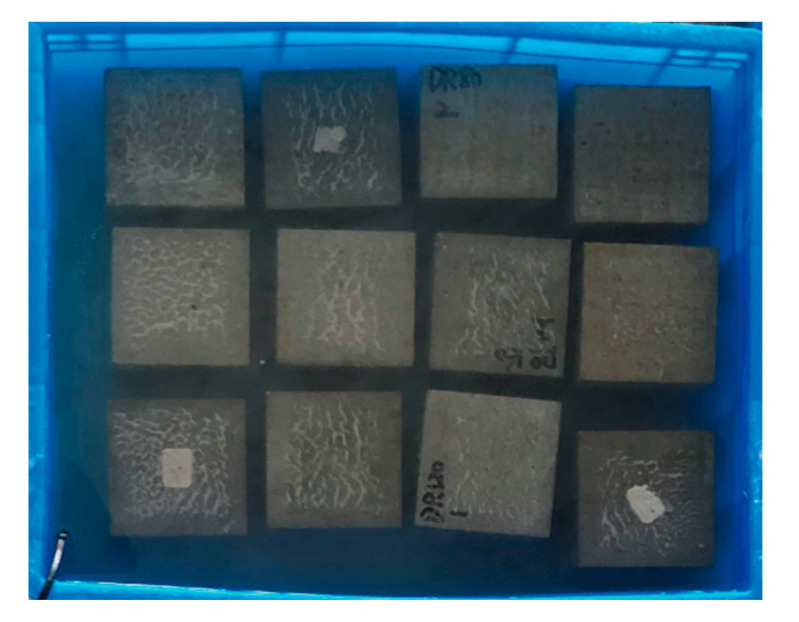
Specimen melting process.

**Figure 5 polymers-15-02066-f005:**
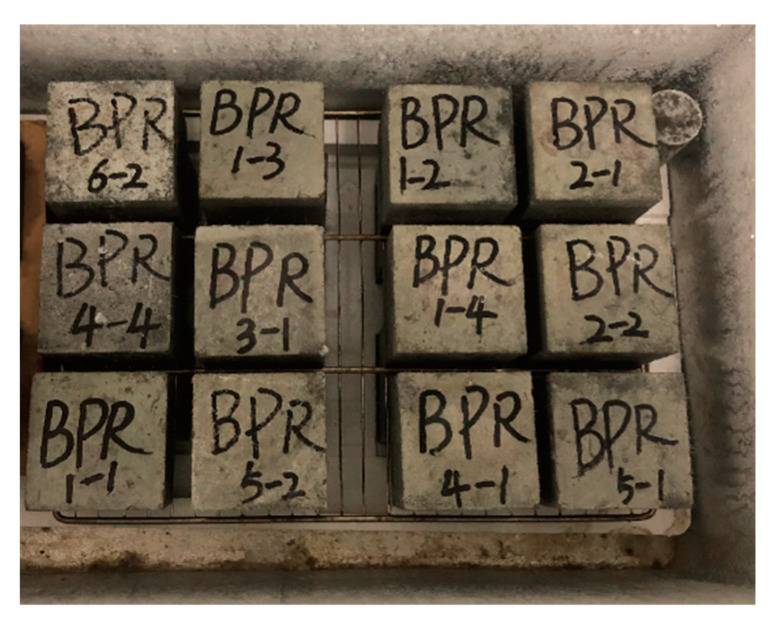
Specimen freezing process.

**Figure 6 polymers-15-02066-f006:**
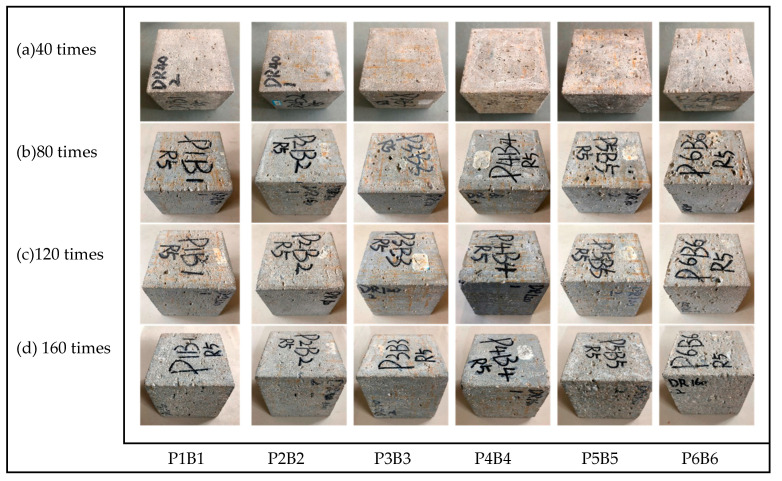
Appearance of PBRC with different contents (in 5% MgSO_4_).

**Figure 7 polymers-15-02066-f007:**
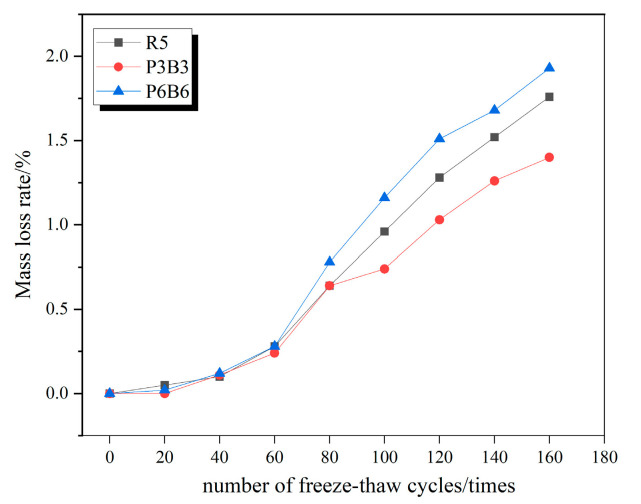
Mass loss rate curve in water.

**Figure 8 polymers-15-02066-f008:**
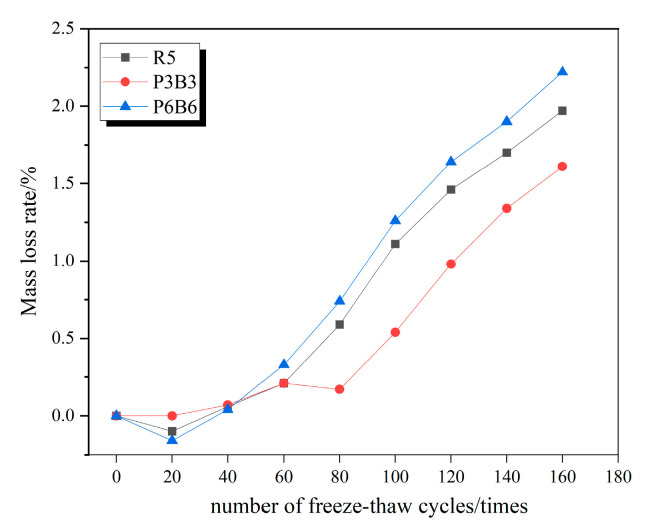
Mass loss rate curve in 5% Na_2_SO_4_.

**Figure 9 polymers-15-02066-f009:**
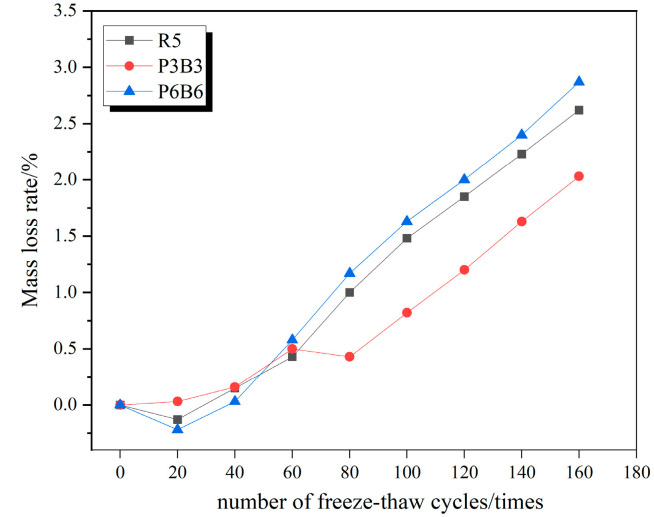
Mass loss rate curve in 5% MgSO_4_.

**Figure 10 polymers-15-02066-f010:**
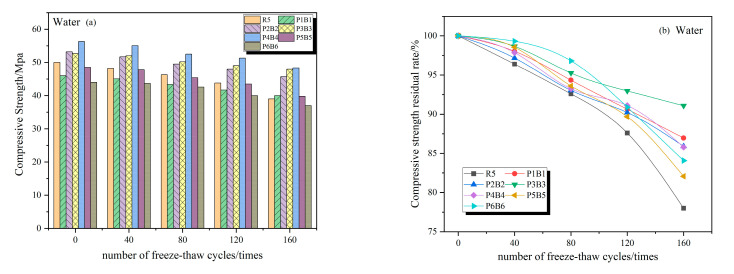
Concrete compressive strength and residual compressive strength. (**a**,**c**,**e**) Concrete compressive strength in water, 5% Na_2_SO_4_, 5% MgSO_4._ (**b**,**d**,**f**) Residual compressive strength in water, 5% Na_2_SO_4_, 5% MgSO_4_.

**Figure 11 polymers-15-02066-f011:**
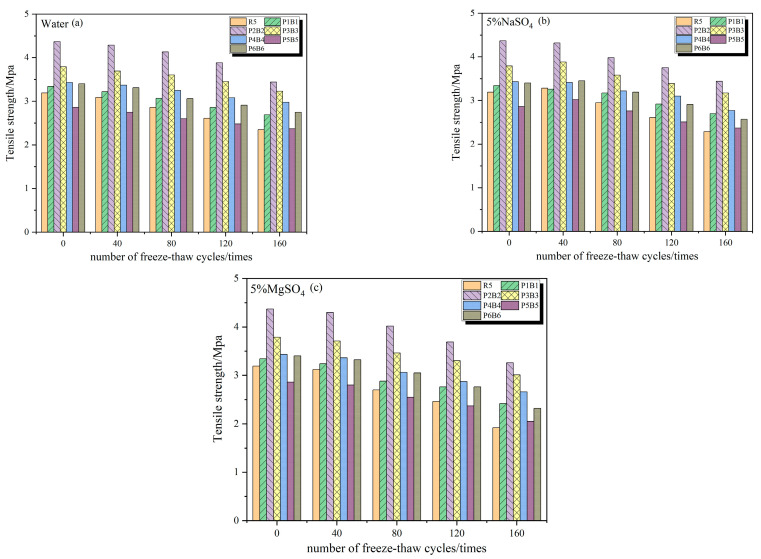
Splitting tensile strength of concrete after freeze–thaw cycles (**a**) in water, (**b**) in 5% Na_2_SO_4_, (**c**) in 5% MgSO_4_.

**Figure 12 polymers-15-02066-f012:**
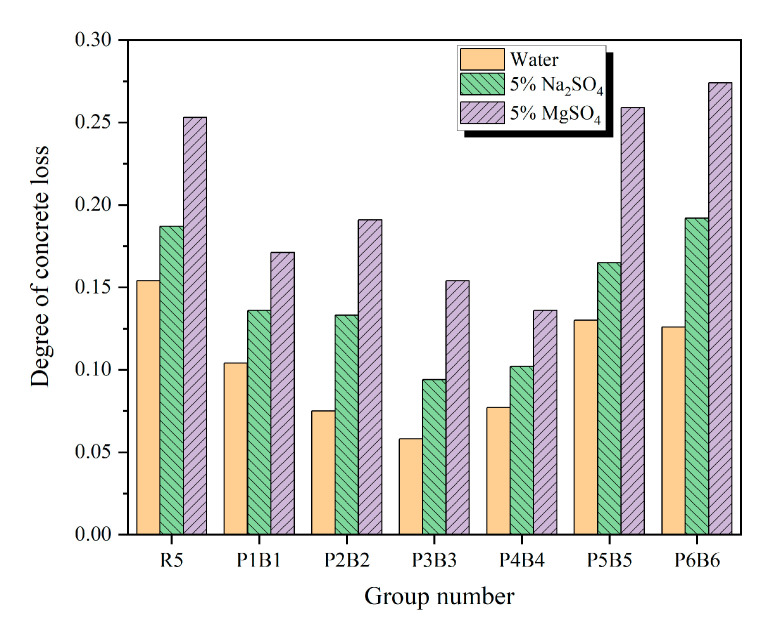
Damage degree of concrete specimens after 160 freeze–thaw cycles with different solutions.

**Figure 13 polymers-15-02066-f013:**
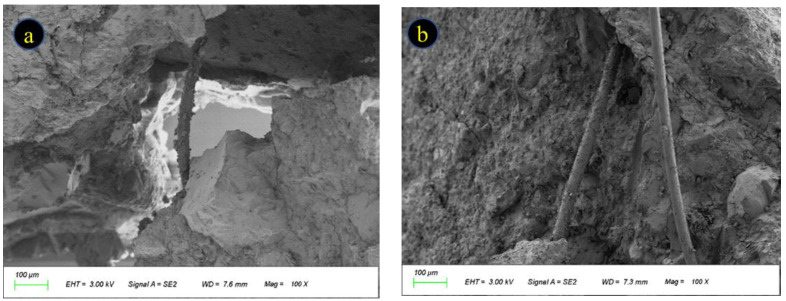
Concrete SEM test results. (**a**,**b**) fiber bridging effect; (**c**,**d**) fiber bonded cement matrix; (**e**,**f**) fiber agglomeration.

**Table 1 polymers-15-02066-t001:** The chemical compositions of cement.

Composition Content (%)	Cement
SiO_2_	22.60
AI_2_O_3_	5.03
Fe_2_O_3_	4.38
CaO	63.11
MgO	1.46
SO_3_	2.24
Loss on Ignition	1.18

**Table 2 polymers-15-02066-t002:** Main performance parameters of fiber.

Fiber Type	Density/ g/cm^3^	Diameter/ µm	Breaking Strength/ MPa	Elastic Modulus/ GPa	Length/ mm
Basalt fiber	2.63	14	4200	96	18
Polypropylene fiber	0.91	26	440	4.8	12

**Table 3 polymers-15-02066-t003:** Base mix ratio of concrete (kg/m^3^).

Material	Cement	Fly Ash	Sand	Limestone	Water	Water Reducer
Content	365	91	602	1107	171	4.5

**Table 4 polymers-15-02066-t004:** Mixing Proportions of the experiment.

Number	Group	BF (%)	PPF (%)	Rubber Content (%)
1	R5	0	0	5
2	P1B1	0.1	0.05	5
3	P2B2	0.2	0.1	5
4	P3B3	0.3	0.15	5
5	P4B4	0.4	0.2	5
6	P5B5	0.5	0.25	5
7	P6B6	0.6	0.3	5

## Data Availability

The data used to support the findings of this study are available from the corresponding author upon request.

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
