# Peer review of "Performance of Rubber Concrete Containing Polypropylene and Basalt Fibers under Coupled Sulfate Attack and Freeze–Thaw Conditions: An Experimental Evaluation"

_polymers, 2023, doi:10.3390/polym15092066_

Round 1

Reviewer 1 Report

This article investigated the response of polypropylene and basalt fiber-reinforced rubber concrete to coupled sulphate attack and freeze-thaw conditions. This research looked at the effects of sulphate attack and freeze-thaw cycles on the morphology, mass, static compressive and tensile tests, ultrasonic non-destructive testing, and scanning electron microscope (SEM) tests. Although the article has been written well, some minor issues need to be addressed.

1. The abstract has mentioned the current problem, objective and methodology. However, the quantitative finding could be added to the abstract.

2. The introduction is adequate and relevant to this study.

3. Adding the granulometric curve for the fine and coarse aggregate is recommended.

4. Please add the cement details and properties in section 2.1. There is no discussion about fly ash in this section.

5. On what basis was the mixing ratio of BF and PPF 2:1was chosen?

6. Why was the content of basalt fiber from 0% to 0.6% selected?.

7. How was the orientation of fibres ensured?

8. The discussion section is clear, and comparing the results with the literature is lacking.

9. It is recommended to add the numerical values in the conclusions.

Author Response

Response to Reviewer #1

This article investigated the response of polypropylene and basalt fiber-reinforced rubber concrete to coupled sulphate attack and freeze-thaw conditions. This research looked at the effects of sulphate attack and freeze-thaw cycles on the morphology, mass, static compressive and tensile tests, ultrasonic non-destructive testing, and scanning electron microscope (SEM) tests. Although the article has been written well, some minor issues need to be addressed.

1.The abstract has mentioned the current problem, objective and methodology. However, the quantitative finding could be added to the abstract.

Thanks for your endorsement and help, the quantitative findings have been added to the Abstract, see lines 18-21.

2.The introduction is adequate and relevant to this study.

Thank you for your approval.

3.Adding the granulometric curve for the fine and coarse aggregate is recommended.

Added particle curves for coarse and fine aggregates on line 194.

4.Please add the cement details and properties in section 2.1. There is no discussion about fly ash in this section.

The chemical composition of the cement used in the experiments is given in Table 1, see row 200 for details.

5.On what basis was the mixing ratio of BF and PPF 2:1was chosen?

We chose this mix ratio with reference to previous research, and a similar mix ratio was mentioned in Song's study (Hybrid Effect Evaluation of Steel Fiber and Carbon Fiber on the Performance of the Fiber Reinforced Concrete https://doi.org/ 10.3390/ma9080704). At the same time, due to the low density of polypropylene fibers, excessive content will also lead to fiber aggregation.

6.Why was the content of basalt fiber from 0% to 0.6% selected?.

Studies have proved that adding 0.45% basalt fiber has a positive effect on the mechanical and microscopic properties of concrete (Efficiency of basalt fiber length and content on mechanical and microstructural properties of hybrid fiber concrete https://doi.org/10.1111/ffe.13483), Therefore, we have refined the range and selected the content of basalt fiber to be 0-0.6%.

7.How was the orientation of fibres ensured?

We hope that the mixed fibers are randomly distributed in the concrete, so that the supporting and bridging functions of the fibers can be fully exerted, and this phenomenon can also be observed in the microscopic images.

8.The discussion section is clear, and comparing the results with the literature is lacking.

Thanks for the suggestion, we have added a comparison with previous studies in the conclusion section, see lines 361-362.

9.It is recommended to add the numerical values in the conclusions.

Thanks for your suggestion, we have revised the conclusions section of the manuscript to add specific numerical values.

Reviewer 2 Report

The authors report on the performance characteristics of hybrid fiber-reinforced rubber concrete in a coupled freeze-thaw erosion environment. Overall, the paper is well written with interesting findings and methods. To improve the clarity of the manuscript, the following points must be addressed:

(1) The abstract should include the problem, objectives, methods, results, conclusions, and recommendations. Please refine the abstract.

(2) Grammatical and punctuation errors left in the manuscript. Please check and correct them.

(3) There is a misplaced letter mark in Figure 9 and Figure 10, place them in the correct position

(4) Why did the authors choose such a fiber mix ratio

(5) Fibers mainly act as bridging and stress transfer in concrete, authors should cite references to prove this effect, such as Influences of MgO and PVA fiber on the abrasion and cracking resistance, pore structure and fractal features of hydraulic concrete; Comparison of fly ash, PVA fiber, MgO and shrinkage-reducing admixture on the frost resistance of face slab concrete via pore structural and fractal analysis

(6) The conclusion section could be more concise, revise this section

(7) The authors should add a statement of interest at the end of the article

Recommended for publication after revision.

Author Response

Response to Reviewer #2

The authors report on the performance characteristics of hybrid fiber-reinforced rubber concrete in a coupled freeze-thaw erosion environment. Overall, the paper is well written with interesting findings and methods. To improve the clarity of the manuscript, the following points must be addressed:

1.The abstract should include the problem, objectives, methods, results, conclusions, and recommendations. Please refine the abstract.

Thanks for your suggestion, we have revised the abstract to add quantitative data.

2.Grammatical and punctuation errors left in the manuscript. Please check and correct them.

We checked the manuscript for grammatical and punctuation errors and made corrections.

3.There is a misplaced letter mark in Figure 9 and Figure 10, place them in the correct position

Thanks for your suggestion, we fixed the errors that appeared in the figure.

4.Why did the authors choose such a fiber mix ratio

We chose this fiber mix ratio with reference to the findings of previous researchers, and a similar mix ratio was mentioned in Song's study (Hybrid Effect Evaluation of Steel Fiber and Carbon Fiber on the Performance of the Fiber Reinforced Concrete https://doi .org/10.3390/ma9080704.

5.Fibers mainly act as bridging and stress transfer in concrete, authors should cite references to prove this effect, such as Influences of MgO and PVA fiber on the abrasion and cracking resistance, pore structure and fractal features of hydraulic concrete; Comparison of fly ash, PVA fiber, MgO and shrinkage-reducing admixture on the frost resistance of face slab concrete via pore structural and fractal analysis

Thank you for your suggestion, the literature has been cited in the manuscript, see lines 56-59 for details.

6.The conclusion section could be more concise, revise this section

Thanks for your help and suggestions, we have revised the Conclusions section as detailed in the revised manuscript.

7.The authors should add a statement of interest at the end of the article

We have added a statement of interest at the end of the article.

Reviewer 3 Report

1. In abstract, due to deficiencies in the mechanics and durability of rubber concrete, it is often reinforced by the addition of fibers. Rewrite this sentence, you probably meant lack of research on the mechanics and durability of rubber concrete.

2. In Introduction, Concrete is one of the most widely used manufacturing materials in the world, use construction materials instead of manufacturing materials.

3. In conclusion, For the same specimen, under a certain number of freeze-thaw cycles, the apparent damage is most serious, replace the damage word with degradation or deterioration.

4. How did you conclude on the spalling of cement paste and fine aggregate on the surface? Refer this finding with the results section of the study.

Author Response

Response to Reviewer #3

1.In abstract, due to deficiencies in the mechanics and durability of rubber concrete, it is often reinforced by the addition of fibers. Rewrite this sentence, you probably meant lack of research on the mechanics and durability of rubber concrete.

Thanks for your suggestion, we have rewritten this passage, see lines 11-12 for details.

2.In Introduction, Concrete is one of the most widely used manufacturing materials in the world, use construction materials instead of manufacturing materials.

We have revised in the manuscript, specifically see lines 30-31.

3.In conclusion, For the same specimen, under a certain number of freeze-thaw cycles, the apparent damage is most serious, replace the damage word with degradation ordeterioration.

Thank you for your suggestion, we have made modifications in the text, see lines 364-366 for details.

4.How did you conclude on the spalling of cement paste and fine aggregate on the surface? Refer this finding with the results section of the study.

We judge the exfoliation of slurry and aggregate by taking photos for comparison and weighing the mass of the test block. The related results are mentioned in the conclusion part, see lines 366-368.
